# Decreased susceptibility to viscosin in *Streptococcus pneumoniae*

Anja Ruud Winther,[1] Zhian Salehian,[1] Cathrine Arnason Bøe,[2] Malene Nesdal,[1] Leiv Sigve Håvarstein,[1] Morten Kjos,[1] Daniel Straume[1]

**ABSTRACT** Growing numbers of infections caused by antibiotic-resistant *Streptococcus pneumoniae* strains are a major concern for healthcare systems that will require new antibiotics for treatment as well as preventative measures that reduce the number of infections. Lipopeptides are antimicrobial molecules, of which some are used as antibiotics, including the last resort antibiotics daptomycin and polymyxins. Here we have studied the antimicrobial effect of the cyclic lipopeptide viscosin on *S. pneumoniae* growth and morphology. Most lipopeptides function as surfactants that create pores in membrane layers, which is regarded as their main antimicrobial activity. We show that viscosin can inhibit growth of *S. pneumoniae* without disintegration of the cytoplasmic membrane. Instead, the cells developed abnormal shapes and misplaced new division sites. The cell wall of these bacteria appeared less dense in electron microscopy images, suggesting that viscosin interfered with normal cell wall synthesis. Corroborating this observation, a luciferase reporter assay was used to show that the two-component systems LiaFSR and CiaRH, which are known to be activated upon cell wall stress, were strongly induced by viscosin. Furthermore, a mutant displaying 1.8-fold decreased susceptibility to viscosin was generated by sequential exposure to increasing concentrations of the lipopeptide. The mutant suffered from significant fitness loss and had mutations in genes involved in fatty acid synthesis, teichoic acid synthesis, and cell wall synthesis as well as transcription and translation. How these mutations might be linked to decreased viscosin susceptibility is discussed.

**IMPORTANCE** *Streptococcus pneumoniae* is a leading cause of bacterial pneumonia, sepsis, and meningitis in children, and the incidence of infections caused by antibiotic-resistant strains is increasing. Development of new antibiotics is therefore necessary to treat these types of infections in the future. Here, we have studied the activity of the antimicrobial lipopeptide viscosin on *S. pneumoniae* and show that in addition to having the typical membrane destabilizing activity of lipopeptides, viscosin inhibits pneumococcal growth by obstructing normal cell wall synthesis. This suggests a more specific mode of action than just the surfactant activity. Furthermore, we show that *S. pneumoniae* does not easily acquire resistance to viscosin, which makes it a promising molecule to explore further, for example, by synthesizing less toxic derivates that can be tested for therapeutic potential.

**KEYWORDS** *Streptococcus pneumoniae*, viscosin, lipopeptide, *Pseudomonas*, antimicrobial activity

Address correspondence to Daniel Straume, daniel.straume@nmbu.no.

The authors declare no conflict of interest.

See the funding table on p. 12.

Antimicrobial-resistant bacteria are a worldwide threat that cause a huge death toll and economic burden on our society. It is estimated that by the year 2050, at least 10 million people will die each year from infections by drug-resistant pathogens (1). *Streptococcus pneumoniae*, also known as pneumococcus, is an important human pathogen that can cause severe sepsis, bacterial pneumonia, and meningitis. Children, elderly, and immunocompromised individuals are particularly susceptible to infections

by the pneumococcus (2). Infections are normally treated successfully with penicillin; however, pneumococcal strains resistant to penicillin and other classes of antibiotics (macrolides, fluoroquinolones, and tetracyclines) are becoming increasingly common (3). When the available antimicrobial agents are no longer useful, development of new compounds with antimicrobial properties is necessary to keep the standard treatments of modern medicine.

Viscosin is a well-known antimicrobial cyclic lipopeptide targeting several bacteria including species within the genera *Streptococcus*, *Mycobacterium, Enterococcus*, and *Clostridium* (4–6). It has also been reported to have activity against fungi, protozoa, and human viruses (4, 6, 7). Viscosin was first discovered in 1950 by the Kochi group (Mutsuyuk Kochi, Yokohama Medical College) (4), and the structure was solved in 1970 by Hiramoto, M. and co-workers. It comprises nine amino acids (L-Leu-D-Glu-D-allo-Thr-D-Val-L-Leu-D-Ser-L-Leu-D-Ser-L-Ile), which are cyclized between the C-terminal hydroxyl group of L-Ile and the side-chain hydroxyl group of D-allo-Thr. L-Leu in position 1 is attached to 3-hydroxydecanoic acid (Fig. 1c) (8). The amphiphilic nature of viscosin is important for its antimicrobial activity, which is considered to involve pore formation and destabilization of the cytoplasmic membrane of target cells (5, 9, 10). Despite having this membrane-targeting activity, several bacterial species are non-susceptible (6) to viscosin (e.g., *Escherichia coli*, *Bacillus cereus*, and *Staphylococcus epidermidis*), which suggests that the antimicrobial activity of viscosin depends on interaction with one or more specific target molecules (e.g., membrane lipids and membrane or surface proteins) that are lacking or structurally different in the cell envelopes of non-susceptible bacteria. However, our understanding of why some bacteria are sensitive to viscosin while others are not remains limited.

In this study, we screened microorganisms sampled from the shore of the Oslo fjord in Norway for production of compounds active against the pneumococcus. One *Pseudomonas* sp. isolate displayed significant inhibition of *S. pneumoniae*, and the compound produced was identified to be viscosin. We show that viscosin can inhibit growth of *S. pneumoniae* without disrupting the integrity of the cytoplasmic membrane. These cells displayed heterogenous sizes and misplacement of septa. Viscosin treatment also activated two-component systems related to cell wall stress (LiaFSR and CiaRH) (11–13). Furthermore, a mutant displaying ~1.8 times decreased susceptibility to viscosin was generated. The mutant, which showed a significant fitness cost, had mutations in genes involved in lipid synthesis, transcription, translation, cell wall, and teichoic acid biosynthesis. The relationship between the mutated genes and decreased susceptibility to viscosin is discussed.

## RESULTS

### Purification of viscosin

Many of the most successful antibiotics in use have been isolated from natural sources (14). In a screen for microorganisms producing antimicrobials against *S. pneumoniae*, we isolated a *Pseudomonas* sp. from the shore of the Oslo fjord just south of Drøbak in Norway (Fig. S1). *Pseudomonas* spp. are known to synthesize a vast number of antimicrobial lipopeptides (15, 16), and we therefore suspected the antimicrobial compound to belong to this family of biomolecules. To uncover which lipopeptide the isolate produced, we first established a purification protocol as depicted in Fig. 1a. Lipopeptides are typically amphiphilic molecules with a hydrophobic hydrocarbon chain linked to a hydrophilic peptide chain. The hydrophobic part of lipopeptides can be exploited for purification purposes in liquid–liquid extractions with organic solvents (17–19). The antimicrobial compound was successfully extracted from heat-treated (100°C) *Pseudomonas* sp. culture supernatants by using *n*-butanol. After removal of *n*-butanol by evaporation, the compound was further purified using C18 reverse phase high performance liquid chromatography (HPLC) (Fig. 1b). The material that eluted as one single peak after ~17 minutes (at approximately 90% acetonitrile) showed antimicrobial activity against *S. pneumoniae*, and it was inactivated after treatment with polymyxin acylase

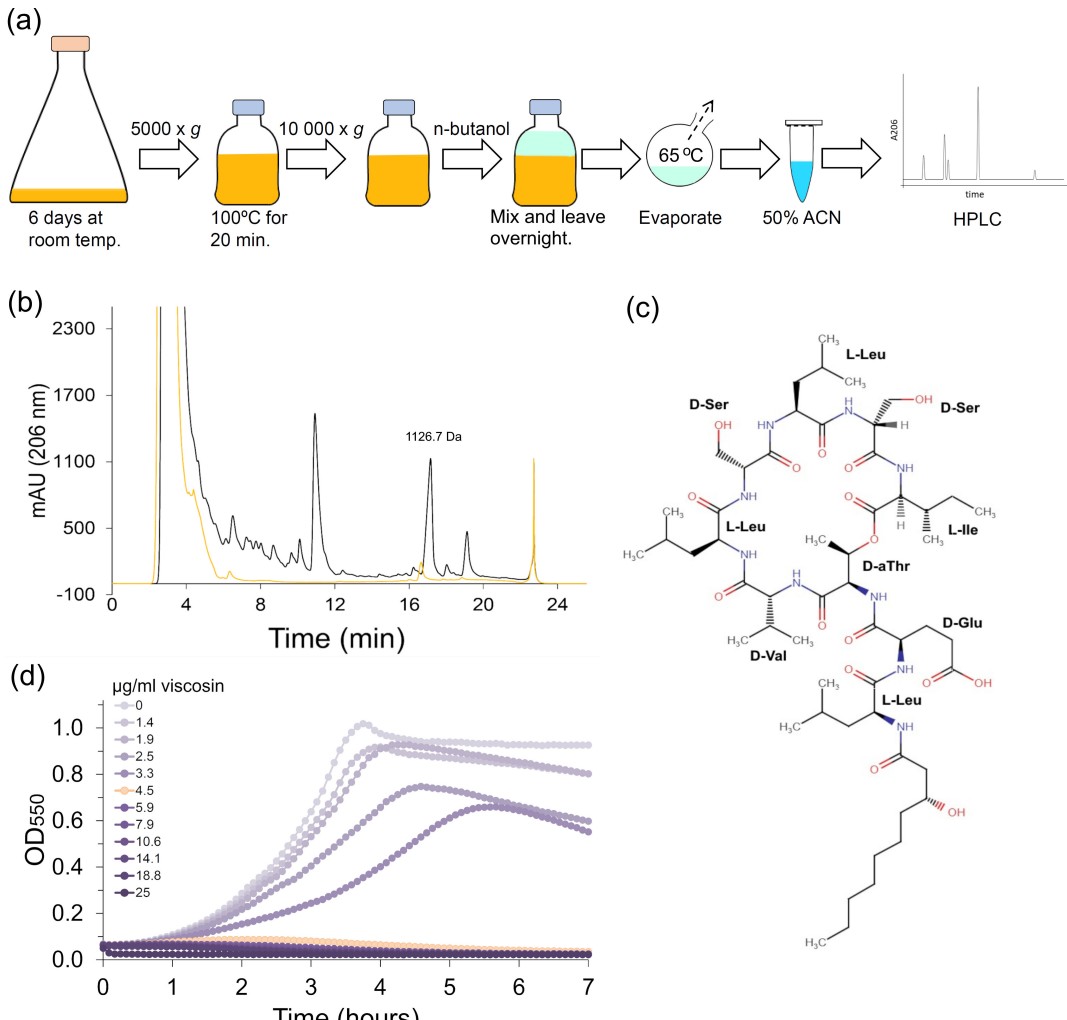

**FIG 1** Purification of viscosin. (a) Scheme illustrating viscosin production, enrichment, and purification. Viscosin was extracted from culture supernatants of *Pseudomonas* sp. grown for 6 days at room temperature using *n*-butanol. After *n*-butanol removal, the dried material was dissolved in acetonitrile/0.1% formic acid (50/50) and separated by reverse phase HPLC. (b) Viscosin eluted at approximately 90% acetonitrile (~17 minutes). Black curve represents the *n*-butanol phase and orange curve the water phase after extraction (c) Chemical structure of viscosin, which is available in the Global Substance Registration System database. The structure was drawn using Chemical Sketch Tool (https://www.rcsb.org/chemical-sketch). (d) *S. pneumoniae* grown in the presence of a 1.3-fold dilution series of viscosin starting at 25 µg/mL. The curve representing more than 90% growth inhibition is colored orange (4.5 µg/mL viscosin).

(Fig. S2), which cleaves the acyl bond between the peptide part and the fatty acid moiety of lipopeptides. Molecular weight analysis of this sample by mass spectrometry detected ion signals at *m/z* 1,126.7, 1,148.7, 1,164.7, and 1,170.7 Da (Fig. S3a) of which the signal at 1,148.7 Da was the most intense. The first three signals (1,126.7, 1,148.7, and 1,164.7 Da) correlated with the weight of the protonated molecular ion and sodium and potassium adducts of viscosin (1,126.4 Da) (20), which is a well-known cyclic lipopeptide produced by species in the *Pseudomonas* genus4, 21, 22). Mass spectrometry in the negative mode confirmed this by detecting an ion signal at *m/z* 1,124.7 Da assigned to the $(M–H)^-$ ion (Fig. S3a). The viscosin obtained in the present work did not contain significant impurities of sizes below 3,500 Da (Fig. S3b), and considering that it eluted from the C18 column at ~90% acetonitrile, we reasoned that the lipopeptide was obtained at >90% purity. The dry weight of purified viscosin was therefore estimated for use in MIC-assays. The $MIC_{90}$ of viscosin against our laboratory *S. pneumoniae* strain (R6 derivate) was estimated to ~4.5 µg/mL as determined in liquid cultures (Fig. 1d).

## Viscosin causes morphological abnormalities and autolysis in *S. pneumoniae*

Viscosin has been shown to inhibit the growth of several Gram-positive bacteria and species of mycobacteria (5, 23). Like other cyclic lipopeptides of the viscosin group [pseudodesmin A, viscosinamide A, and white line-inducing principle (WLIP)], it interacts with membrane lipids (5, 9). This interaction is recognized as a key factor for their antimicrobial activity. To check if viscosin kills *S. pneumoniae* by interfering with the cell membrane integrity, we used the DNA-binding dye SYTOX Green. It fluoresces when bound to DNA but is unable to cross intact cell membranes. When adding ~2× $MIC_{90}$ of viscosin (10 µg/mL) to exponential growing pneumococci, the fluorescent signal from SYTOX Green increased rapidly (first measurement 5 minutes after addition of viscosin), and cell lysis was also observed (Fig. 2c). This shows that higher concentrations of viscosin disintegrated the membrane, giving SYTOX Green access to DNA, and that the activity of the autolysin LytA was induced. Exponentially growing pneumococci are normally immune to LytA (24, 25) but become LytA-sensitive when the integrity of the cell membrane is disturbed, for example by detergents such as Triton X-100 or deoxycholate (25–27). In contrast, when treating cells with concentrations of viscosin just below (4 µg/mL) and above (6 µg/mL) the $MIC_{90}$ value, most cells had intact membranes (Fig. 2a). However, they started to grow in short chains and developed heterogenous sizes (Fig. 2a and b). The average cell size increased from 0.72 ± 0.18 µm$^2$ (untreated) to 0.99 ± 0.39 µm$^2$ and 0.90 ± 0.41 µm$^2$ for cells treated with 4- and 6 µg/mL viscosin, respectively. Staining of the nucleoids with 4′,6-diamidino-2-phenylindole (DAPI) showed uneven staining for treatment with 6 µg/mL viscosin, and anucleated cells were observed for both viscosin concentrations (Fig. 2a). Although 6 µg/mL is ~30% more viscosin than the $MIC_{90}$ concentration of 4.5 µg/mL, live/dead staining showed that most of the cells contained an intact membrane. This suggests that viscosin can inhibit cell growth without disrupting the cell membrane, which occurs at higher concentrations (10 µg/mL).

## Viscosin induces cell wall stress

The heterogenous cell sizes caused by viscosin suggest that it interferes with septum placement and/or cell wall synthesis. This was confirmed by transmission electron microscopy (TEM), which showed that cells grown with 4.5 µg/mL viscosin ($MIC_{90}$) for 4 hours contained multiple misplaced septa (Fig. 3a). In addition, the TEM micrographs suggested that the cell wall of treated cells were less dense since they showed lower electron scattering (displayed lower contrast) than the cell wall of non-treated cells. Cell wall stress is known to activate genes controlled by three different two-component systems called WalRK, LiaFSR, and CiaRH in *S. pneumoniae* (11–13, 28, 29). The WalRK is essential and controls among other genes several that encode peptidoglycan hydrolases involved in cell division as well as LysM-domain proteins (peptidoglycan binding) (28, 30). The LiaFSR and CiaRH systems are non-essential and activate genes of various functions (phage-shock proteins, proteases, teichoic acid biosynthesis, sugar uptake, and cell wall polysaccharide metabolism) in response to cell wall stress (11, 31, 32). The molecular mechanisms by which these systems protect against stress to the cell wall are largely unknown although a phage-shock-like protein response has been suggested for LiaFSR in *Bacillus subtilis* (33, 34). If viscosin somehow causes cell wall stress, we would

**TABLE 1** Mutations found in the genome of strain ds971

| Affected gene/locus | [a]Mutation | Position on ref. genome (NC_003098.1) | Effect |
|---|---|---|---|
| *fabK* | c.647T > C | 379097 | p.I216T |
| *fakB3/acrR* | g.659429_659430insT | 659429_659430 | Promoter |
| *rpsA* | c.853_854insAA | 760589_760590 | p.T285Kfs*6 |
| *rpoB* | c.1327G > A | 1750345 | p.R443C |
| *murZ* | c.253_254insTCCAATGCCTT | 974048_974049 | p.Y85Ffs*22 |
| *dltD* | c.1124_1127del | 1969856_1969853 | p.L378Vfs*24 |

[a]Nomenclature based on the recommendations of human genome variation society.

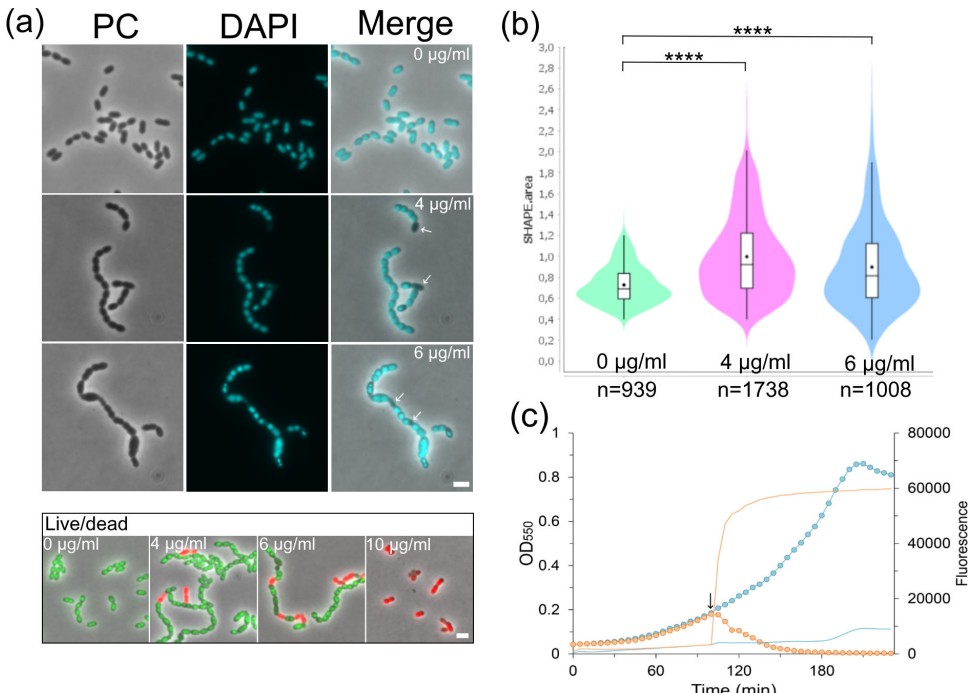

**FIG 2** Phenotypic effects of viscosin on *S. pneumoniae*. (a) *S. pneumoniae* strain RH14 (Δ*lytA*) treated with 4- and 6 µg/mL viscosin (1.125 fold below and 1.3 fold above MIC$_{90}$, respectively) for 4 hours formed short chains, heterogenous cell sizes, and anucleated cells (arrows). Live/dead staining showed that the membrane integrity was completely lost at 10 µg/mL viscosin. Scale bars, 2 µm. (b) Untreated RH14 cells had an average cell size of 0.72 ± 0.18 µm$^2$ compared to 0.99 ± 0.39 µm$^2$ and 0.90 ± 0.41 µm$^2$ for cells treated with 4- and 6 µg/mL viscosin, respectively. *P* values were obtained relative to wild type using one-way analysis of variance, ****$P$ < 0.0001. (c) Growth assay of strain RH425 (LytA$^+$) showing viscosin-induced autolysis. Circled curves represent OD$_{550}$, while non-symbol curves represent SYTOX Green fluorescence (blue curves are non-treated cells). Addition of 10 µg/mL viscosin cell culture at OD$_{550}$ = 0.2 (black arrow) induced autolysis of exponentially growing *S. pneumoniae*. Loss of membrane integrity was measured as increase in fluorescence (485/535 nm).

expect at least one of these systems to become activated. To test this, we used three strains in which expression of the reporter gene *luc* (encoding firefly luciferase) was regulated by each system (Fig. 3b). The *luc* gene was expressed from the promoter of *pcsB* [WalRK regulon (35)], *htrA* [CiaHR regulon (31)], and *liaF* [LiaFSR regulon (11)]. The production of luciferase increased in a dose-dependent manner with increasing concentrations of viscosin for both the P$_{liaF}$ and P$_{htrA}$ promoters, but not for the P$_{pcsB}$ promoter (Fig. 3b). The P$_{pcsB}$ promoter was, however, induced earlier during growth with higher viscosin concentrations. In sum, these results indicate that viscosin somehow causes stress to the cell wall also at sublethal concentrations.

## Decreased susceptibility to viscosin

To get clues about the molecular mechanism by which viscosin inhibits pneumococcal growth, we tried to generate a viscosin-resistant mutant and identify the resistance-conferring mutations. *S. pneumoniae* was cultivated with increasing viscosin concentrations starting at 5 µg/mL. After 24–72 hours, visible growth was observed, and the cells were re-diluted in fresh medium containing 1 µg/mL more viscosin. This procedure was repeated until growth was observed in medium containing 20 µg/mL viscosin (Fig. 4a). A Δ*lytA* strain (RH14) was used for this purpose to avoid autolysis. The cells that had grown for 72 hours with 20 µg/mL viscosin displayed long chains of more than 50 cells. However, most were dead (loss of membrane integrity) and lacked DNA (Fig. 4a). A pure culture of the mutant, which was named strain ds971, was obtained by re-streaking on Todd Hewitt (TH) agar. The MIC$_{90}$ of ds971 was estimated to 10.6 µg/mL compared to

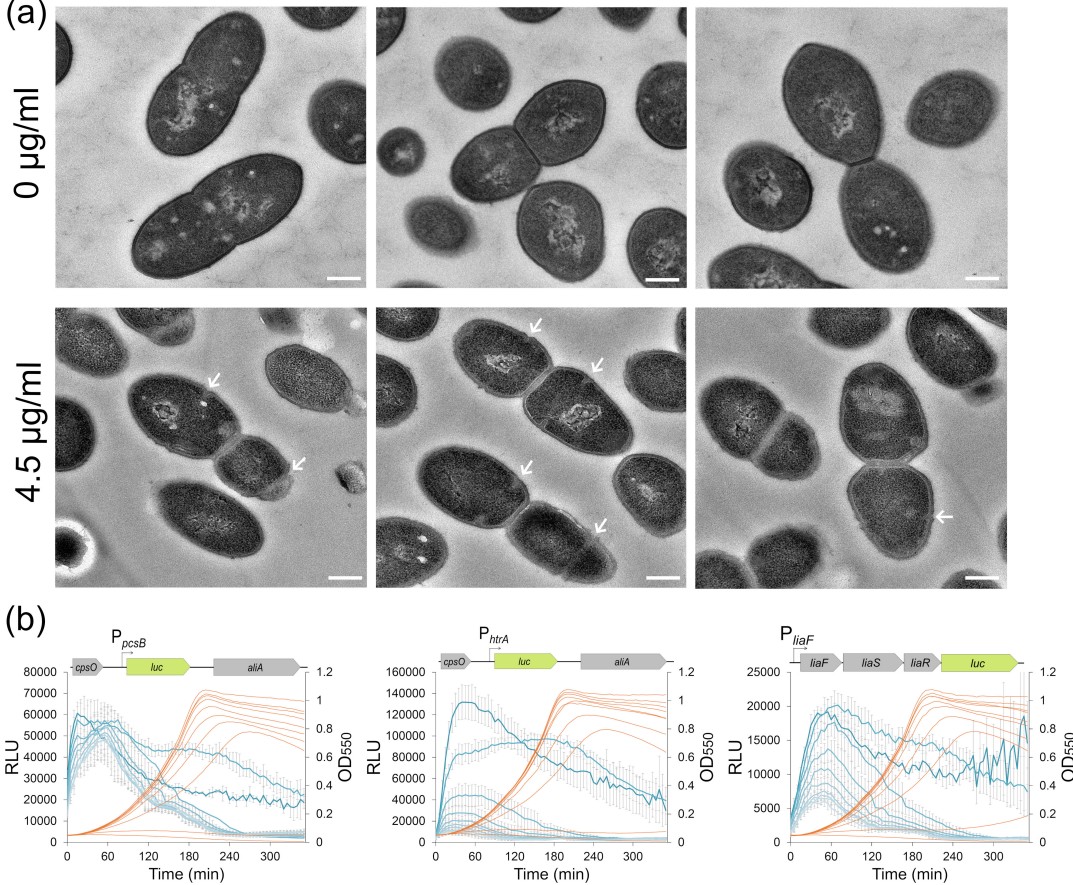

**FIG 3** Viscosin induces cell wall stress in *S. pneumoniae*. (a) Transmission electron micrographs of *S. pneumoniae* treated with 0- and 4.5 µg/mL viscosin for 4 hours. Cells treated with viscosin contained misplaced septa (arrows), and the peptidoglycan layer scattered less electrons (less contrast) compared to non-treated cells. Scale bars, 200 nm. (b) Luciferase reporter assay measuring the promoter activity of $P_{pcsB}$ (WalRK system), $P_{htrA}$ (CiaRH system), and $P_{liaF}$ (LiaFSR system) in *S. pneumoniae* upon exposure to increasing concentrations of viscosin (diluted 1.3-fold starting at 8.4 µg/mL). Orange curves represent $OD_{550}$, and blue curves are relative luminescence units.

5.9 µg/mL for the parental RH14 strain, which corresponded to a 1.8-fold increase (Fig. 4b). The decrease in susceptibility came with a fitness cost, since ds971 grew significantly slower than RH14, and the cell sizes were highly heterogenous. Nevertheless, live/dead staining suggested that most cells were viable (Fig. 4c). The long chains initially observed for cells grown with 20 µg/mL viscosin were caused by the presence of viscosin, and not an intrinsic property of the mutant since strain ds971 did not form long chains under normal growth conditions but started to grow in chains when viscosin was added for 18 hours (Fig. 4c). This morphology was reversible upon viscosin removal.

Genome sequencing of strain ds971 showed that it had acquired multiple mutations. Mutations were identified in genes involved in fatty acid synthesis (*fabK*), transcription (*rpoB*), translation (*rpsA*), and cell wall synthesis (*murZ* and *dltD*). In addition, one mutation was found in the promoter region of *fakB3* (DegV family protein, putative fatty acid binding) and *acrR* (TetR family, transcriptional regulator). The mutations are summarized in Table 1. The mutations in *fabK* and *rpoB* resulted in single amino acid substitutions (FabK$^{I216T}$ and RpoB$^{R443C}$), whereas the mutations in *rpsA*, *murZ*, and *dltD* resulted in codon frame shifts and premature termination of translation (RpsA$^{T285Kfs*6}$, MurZ$^{Y85Ffs*22}$, and DltD$^{L378Vfs*24}$).

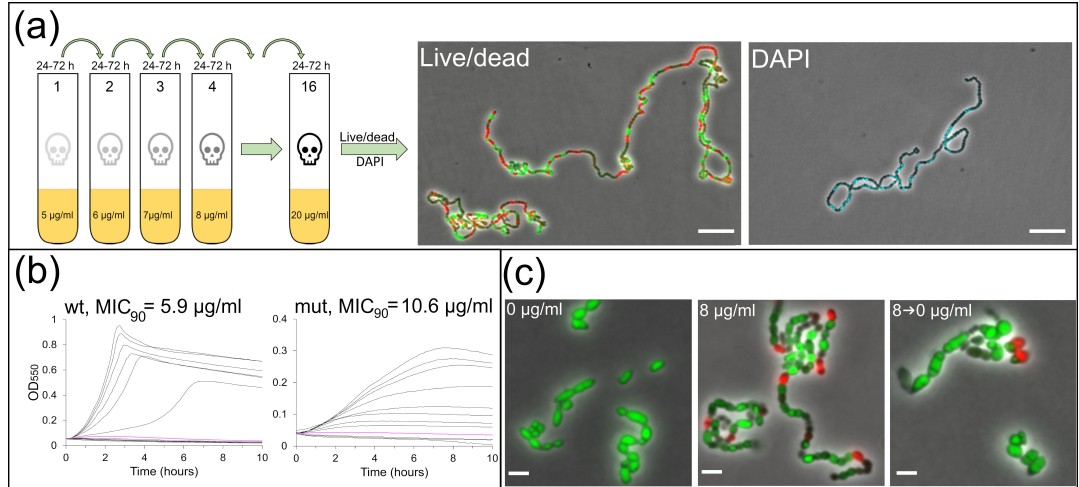

**FIG 4** Decreased susceptibility to viscosin comes with a fitness cost. (a) *S. pneumonaie* RH14 (Δ*lytA*) was repeatedly inoculated in growth medium containing 1 µg/mL higher concentration of viscosin than the previous culture. By starting at 5 µg/mL, a mutant growing at 20 µg/mL (after 16 re-inoculations) was obtained (strain ds971). Cells grown for ~72 hours with 20 µg/mL viscosin formed long chains containing both live and dead bacteria. Several cells in a chain lacked DNA. Scale bars, 5 µm. (b) The $MIC_{90}$ of strain ds971 was ~1.8 times higher than strain RH14, and the growth was reduced. (c) Strain ds971 displayed heterogenous cell sizes. Chain formation was induced by addition of 8 µg/mL viscosin for 18 hours, which was reversible upon viscosin removal. Scale bars, 2 µm.

## DISCUSSION

Lipopeptides are effective antimicrobial molecules targeting a broad range of pathogens (36, 37). In addition, acquisition of resistance to lipopeptides seems to be relatively rare (38–40), making them attractive for antibiotic development. Here, we have studied the cyclic lipopeptide viscosin, which shows strong antimicrobial activity against the pneumococcus. We explored the phenotypic changes happening to the pneumococcus upon viscosin exposure and whether resistance to this lipopeptide can be acquired by growing *S. pneumoniae* with increasing concentrations of the compound over a ~5-week time period. Studies have shown that viscosin functions as a biosurfactant that perturbates membrane layers, and pore formation in cytoplasmic membranes is therefore believed to be the main antimicrobial mechanism of action (5, 6, 10). The exact nature of these pores is, however, poorly characterized. We confirmed that a viscosin concentration corresponding to ~2× $MIC_{90}$ (10 µg/mL) kills *S. pneumoniae* by disrupting the membrane integrity (Fig. 2). In addition, our data suggest that viscosin at lower concentrations (4 µg/mL and 6 µg/mL) efficiently inhibited pneumococcal growth without disintegrating the cytoplasmic membrane (Fig. 2). Most cells treated with 4- (0.9× $MIC_{90}$) and 6 µg/mL (1.3× $MIC_{90}$) viscosin had their cytoplasmic membrane intact. Instead, cells obtained irregular sizes with misplaced septa, started to grow in chains,

**TABLE 2** Bacterial strains used in this work

| Strains | Genotype | Source |
|---|---|---|
| *S. pneumoniae* R6 derivates | | |
| RH425 | Δ*comA*::*ermAM, rpsL1*; Ery[r], Sm[r] | (78) |
| RH14 | Δ*comA*::*ermAM*, Δ*lytA*::kan; Ery[r], Kan[r] | (26) |
| RH259 | Δ*comA*::*ermAM, rpsL1,* Δ*bgaA*::*spc*, P*liaF*::luc; Ery[r], Sm[r], Spc[r], Cm[r] | (11) |
| aw264 | Δ*comA*::*ermAM, rpsL1,* Δ*spr0324*::Janus; Ery[r], Kan[r] | This work |
| aw369 | Δ*comA*::*ermAM, rpsL1,* Δ*spr0324*::P*htrA*-*luc*; Ery[r], Sm[r] | This work |
| SPH127 | Δ*comA*::*ermAM, rpsL1,* P*comX*-*luc,* Δ*spr0324*::Janus; Ery[r], Kan[r] | (79) |
| SPH261 | Δ*comA*::*ermAM, rpsL1,* Δ*IS1167*::P1-*comR,* Δ*spr0324*::P*pcsB*-*luc*; Ery[r], Sm[r] | (80) |
| ds971 | Mutations in *fabK, fakB3, rpsA, rpoB, murZ* and *dltD* (see Table 1 for details). | This work |
| Other strains | | |
| *Pseudomonas* sp. | Wild type, viscosin producer | Lab collection |

and some were anucleated, showing that the cell division machinery malfunctions at these viscosin concentrations. In line with this result, the cell wall stress indicator systems LiaFSR and CiaHR were strongly induced by viscosin, and the cells appeared to be enveloped by a cell wall of less density than normal (Fig. 3). The latter has also been reported for pneumococci treated with moenomycin, which inhibits the glycosyl transferase activity of Class A penicillin binding proteins (PBPs) (41). Together, these data suggest that viscosin somehow interferes with normal cell wall synthesis. It has been shown that some lipopeptides act as antimicrobials by blocking components of the cell wall biosynthesis pathway. For example, daptomycin forms a complex with bactoprenyl-coupled lipid intermediates and phosphatidylglycerol to inhibit peptido-glycan synthesis (42), and amphomycin blocks MraY (phospho-MurNAc-pentapeptide translocase) which transfers the peptidoglycan precursor phospho-MurNAc-pentapep-tide to the lipid undecaprenyl phosphate resulting in lipid I (43, 44). We found that daptomycin treatment of *S. pneumoniae* resulted in multiple incomplete septa, but not cell walls of lower density (Fig. S4), as observed for viscosin-treated cells, suggesting that viscosin has a different mode of action. Viscosin could potentially use other components involved in cell wall synthesis as docking molecule for insertion into the membrane. Alternatively, it is possible that viscosin is inserted into the membrane without significant pore formation but instead alters the membrane compartmentalization, which could interfere with the organization of membrane proteins involved in cell wall synthesis (45). It should be noted that the live/dead and SYTOX Green assays used in this work do not detect pores allowing passage of molecules smaller than 278 Da. Hence, we cannot exclude that cytoplasmic membranes of cells treated with concentrations of viscosin lower than 10 µg/mL are permeable to smaller solutes such as $H^+$ and $K^+$ ions leading to destabilization of the membrane potential and eventually cell death. In fact, it has been shown that loss of membrane potential leads to mislocalization of cell division proteins (46), which could explain the observed morphological changes.

Although resistance to lipopeptides is not easily acquired (38–40), resistance to lipopeptides in clinical use, that is, daptomycin, polymyxin B and E, caspofungin, and micafungin, has been reported (47–52). To examine if *S. pneumoniae* could acquire resistance to viscosin, we treated a Δ*lytA* mutant (to avoid autolysis) with increasing concentrations of viscosin for 5 weeks. A mutant with 1.8 times higher $MIC_{90}$ than the parental strain was obtained. This shows that resistance to viscosin is not easily acquired by the pneumococcus. Furthermore, the mutant displayed a significant fitness loss (Fig. 4) as a result of mutations in *fabK*, *rpoB*, *rpsA*, *dltD*, and *murZ* and in the promoter region of *fakB3* and *acrR*. Several of the mutated genes and promoters have functions related to the cell envelope (*dltD*, *murZ*, *fabK*, and *fakB3*). The non-sense mutations in *dltD* and *murZ* most probably resulted in complete inactivation of the proteins or reduced their functionality. DltD is required for incorporation of D-alanines into lipo- and cell wall teichoic acids (53, 54), and MurZ (also called MurA2) is one of two MurA homologs in *S. pneumoniae*, which are involved in the first commitment step of peptidoglycan biosynthesis by transferring an enolpyruvyl group from phosphoenolpyr-uvate to UDP-N-acetylglucosamine resulting in UDP-N-acetylglucosamine enolpyruvate (55, 56). Deletion of *murZ* is possible but has been shown to reduce the velocity of septal peptidoglycan synthesis (57). How the mutation in *murZ* contributes to viscosin tolerance is not clear, but it could possibly lead to changes in the cell wall structure that contributes to this. D-Alanine-decorated teichoic acids have been shown to protect bacteria against positively charged antimicrobial peptides by electrostatic repulsion (54, 58). The peptide part of viscosin (LETVLSLSI) contains a glutamic acid (pKa value of side group, ~4), most probably giving the lipopeptide a negative charge at neutral pH in the growth medium. Hence, lack of D-alanine-decorated teichoic acids (*dltD*-mutation) in the ds971 mutant increases the net negative charge on the cell surface which will create a repulsive force making it more difficult for viscosin to access its target in the cytoplasmic membrane. We also found a mutation in the *fabK* gene (missense, I216T) and in the promoter of *fakB3* (insertion). FabK is an essential enoyl-acyl carrier protein,

which elongates fatty acids, while FakB3 [probably not essential (59)] has been shown to bind fatty acids (60, 61). It is possible that mutations affecting the function and/or expression levels of these proteins somehow change the membrane lipid composition in the mutant to better tolerate the surfactant function of viscosin.

The mutations in *rpsA* (non-sense) and *rpoB* (missense, R443C) probably affect both translation and transcription. The *rpsA* gene encodes the ribosomal protein S1 that is shown to be important for initiating translation of mRNAs with secondary structures at the 5′ region, and *rpoB* encodes the β subunit of RNA polymerase (62–64). Mutations in *rpoB* have frequently been observed in *Staphylococcus aureus* with decreased susceptibility to the lipopeptide daptomycin (65–68). Mutations in *rpsA* is linked to pyrazinamide resistance, a drug used for treatment of tuberculosis (69), and it has also been shown to mutate in bacteria stressed with various chemicals (70). Finally, the base insertion in the promoter region of *fakB3* could possibly affect expression of *acrR* located in the opposite direction on the genome. AcrR family proteins regulate a wide range of cellular processes including osmotic stress, drug resistance, efflux pumps, and catabolic pathways (71). It is not clear how mutations in *rpsA*, *rpoB*, and the *acrR* promoter directly contribute to increased tolerance to viscosin, but mutations affecting these genes could result in a global fine-tuning of gene expression which is advantageous upon viscosin exposure similar to *rpoB* in oxacillin-resistant *S. aureus* (72). In *S. aureus*, a missense mutation in *rpoB* resulted in an RNA polymerase with reduced ability to initiate transcription from some promoters; in addition, it displayed abnormal pausing properties on DNA templates which will influence gene expression (72, 73). It is also possible that some of the mutations in strain ds971 do not contribute to decreasing viscosin susceptibility but instead compensate for the fitness cost of mutations in other genes.

Taken together, viscosin is an efficient antimicrobial molecule that kills *S. pneumoniae*, and resistance to this compound is not easily acquired by this bacterium. These are promising characteristics, but further exploration of its potential as a future antibiotic depends on low degree of cytotoxicity. Information about this is, however, rather limited. One study has tested the effect of viscosin on prostate and breast cancer cell migration. They found no toxic effect on prostate cancer cells at 22.5 µg/mL (20 µM) but a toxic effect on breast cancer cells at 16.9 µg/mL (15 µM) (22). Also, a lipopeptide called WLIP, which is structurally similar to viscosin (D-leu instead of L-Leu in Position 5), displayed hemolytic activity on blood agar (74). In comparison, we found that viscosin caused hemolysis at >20 µg/mL (17.7 µM) (Fig. S5). This is ~4.4-fold higher than the MIC$_{90}$ concentrations (4.5 µg/mL) for *S. pneumoniae*, which makes further testing in pre-clinical studies (e.g. mouse models) impractical. Nonetheless, derivatization of natural lipopeptides to reduce cytotoxicity while retaining their antimicrobial activity is possible (75, 76). Further insights into the mode of action of viscosin will probably be required if such an approach should be explored in the future.

## MATERIALS AND METHODS

### Bacterial strains, growth conditions, and transformation

Bacterial species and strains are listed in Table 2. *S. pneumoniae* was grown at 37°C in C medium (77), TH broth (Becton, Dickinson and Company) or anaerobically on TH-agar using an air-tight container containing an AnaeroGen bag from Oxoid. When necessary, a final concentration of 400 µg/mL of kanamycin or 200 µg/mL of streptomycin was added to the growth medium. *Pseudomonas* sp. was grown in brain heart infusion (BHI) broth (Oxoid) at room temperature (23°C–25°C) without shaking. For viscosin production, *Pseudomonas* sp. was grown in batches of 1 L BHI for 6 days in a 3-L Erlenmeyer flask (Corning) with an air vent cap.

*S. pneumoniae* was transformed by natural transformation. OD$_{550}$ = 0.05 cultures were induced to competence by adding a final concentration of 250 ng/mL competence stimulating peptide (CSP-1) (81) along with 200–300 ng of transforming DNA. The cells

were incubated at 37°C for 2 hours before transformants were selected on TH agar containing the appropriate antibiotic.

## DNA techniques

Gene knockout cassettes and gene mutations were constructed using overlap extension PCR as previously described by Johnsborg et al. (82). The overlap PCR technique is based on the work by Higuchi et al. (83). Primers are listed in Table S1. Typically, a gene knockout cassette is constructed by fusing the ~1000-bp regions flanking a target site in the genome to a desired antibiotic resistance gene. When introducing mutations in the genome, the Janus knockout cassette (84), containing a kanamycin resistance gene and an *rpsL*-allele conferring streptomycin sensitivity, was first inserted into the desired site in the genome of strains expressing *rpsL1* (conferring streptomycin resistance). Then, Janus was replaced by a DNA fragment containing mutations of interest. To create strain aw369, *spr0324* in strain RH425 was first replaced by Janus resulting in strain aw264. The Δ*spr0324*::Janus cassette was amplified from the genome of strain SPH127 using primer pair KHB50/KHB53. Then, the Janus in aw264 was replaced with P$_{htrA}$-*luc*. An ~800-bp fragment upstream *spr0324* was amplified with primer pair KHB50/KHB51 using genomic DNA from strain RH425 as template. The P$_{htrA}$ promoter was amplified from RH425 genomic DNA using primer pair aw214/aw215. The *luc* gene including ~750-bp downstream of *spr0324* was amplified with primer pair KHB78/KHB53 using genomic DNA from strain SPH261. The three PCR products were subsequently fused and the resulting P$_{htrA}$-*luc* cassette replaced Janus in strain aw264.

## HPLC purification and detection of viscosin

Dry material of viscosin enriched from *n*-butanol extraction (see results section) was dissolved in 50% (vol/vol) acetonitrile in 0.1% (vol/vol) formic acid. Insoluble materials were removed by centrifugation at 20,000 × *g* for 20 minutes. The dissolved viscosin samples were injected into a Halo 160 Å ES-C18, 2.7 µm, 4.6 × 250 mm column (Advanced Materials Technology) that was pre-equilibrated in 50% acetonitrile in 0.1% formic acid. Viscosin was eluted using a linear gradient of acetonitrile from 50%–100% in 0.1% formic acid. Purified viscosin was lyophilized and dissolved in 25 mM NaCO$_3$ at pH 11 (increased solubility). Then, the pH was adjusted to 8.0 by adding a final concentration of 50 mM Tris-HCl, pH 8.0.

For mass determination, lyophilized viscosin was rehydrated in 0.1% trifluoroacetic acid and desalted by passing the sample through C18 STAGE tips (85). Viscosin was eluted in 10 µL of 50% acetonitrile, and 1 µL was mixed 1:1 with matrix (2,5-dihydroxybenzoic acid). The molecular mass was determined by MALDI-TOF using an UltrafleXtreme MALDI-TOF/TOF from Bruker Daltonics.

## MIC- and SYTOX Green assay

A 1.3-fold dilution series of viscosin from 50 to 2.8 µg/mL was prepared in final volumes of 150 µL C medium in a 96-well microtiter plate. One well containing 150 µL C medium without viscosin was used as control. Pneumococcal strains were grown to OD$_{550}$ = 0.3 before they were diluted in C medium to OD$_{550}$ = 0.1. Volumes of 150 µL cell culture were added to the wells in the microtiter plate already containing 150 µL C medium with or without viscosin, resulting in a final OD$_{550}$ = 0.05 in all wells and the 1.3-fold dilution series of viscosin starting at 25 µg/mL. The plate was incubated at 37°C in a Hidex Sense plate reader, and OD$_{550}$ was measured automatically every 5 minutes.

To measure cell membrane integrity of *S. pneumoniae*, 300 µL of exponentially growing bacteria at OD$_{550}$ = 0.05 was added a final concentration of 2 µM SYTOX Green (Thermo Fisher Scientific) in a 96-well microtiter plate. The plate was incubated as described above, and viscosin was added when OD$_{550}$ reached 0.2. Fluorescence was measured at 485/535 nm.

## Luciferase reporter assay

All strains assayed for $P_{pcsB}$-, $P_{htrA}$-, and $P_{liaF}$-driven *luc* reporter activity were grown in C medium to $OD_{550}$ ~0.3. The bacterial cultures were then diluted to $OD_{550}$ = 0.05 in C medium and transferred to a white 96-well Corning non-binding surface (NBS) clear-bottom plate containing a 1.3-fold dilution series of viscosin resulting in final concentrations ranging from 8.4 to 0.8 µg/mL. Wells without viscosin served as controls. D-Luciferin (Thermo Fisher Scientific) was added to a final concentration of 10 mM. The plate was incubated in a Hidex Sense plate reader at 37°C, and $OD_{550}$ and luminescence were measured automatically every 5 minutes throughout the experiment.

## Microscopic analyses

Phase contrast and fluorescence microscopy were performed on bacteria immobilized on a thin layer (<0.5 mm) of 1.2% (wt/vol) agarose in phosphate-buffered saline (PBS). Pictures were taken by using a Zeiss Axio Observer with ZEN Blue software, an ORCA-Flash 4.0 V2 Digital CMOS camera (Hamamatsu Photonics), and a 100× phase-contrast objective. An HXP 120 Illuminator (Zeiss) served as the light source for fluorescence microscopy. Bacterial nucleoids were stained with a final concentration of 0.2 µg/mL DAPI for 5 minutes prior to imaging. For live/dead staining (Live/Dead *Bac*Light, Thermo Fisher Scientific) 3 µL of a 1:1 mixture of propidium iodide (20 mM) and Syto 9 (3.34 mM) were added to 1 mL cell culture followed by incubation for 15 minutes in the dark before imaging. Images were analyzed using the ImageJ software with the MicrobeJ plugin (86).

For TEM, *S. pneumoniae* RH14 was grown in C medium (for viscosin treatment) or TH broth containing 1.25 mM $CaCl_2$ (for daptomycin treatment) to exponential growth phase ($OD_{550}$ = 0.3) before the cells were diluted to $OD_{550}$ = 0.1. The cultures were split into two 10-mL cultures. One parallel served as control of untreated cells that were harvested at $OD_{550}$ = 0.3. Cells in the other parallel were treated with a final concentration of 4.5 µg/mL viscosin or 0.75 µg/mL daptomycin. After 4 hours of incubation at 37°C, the treated cells were harvested at 4000 × *g*. Cell pellets were re-suspended in fixative solution [2% paraformaldehyde (wt/vol), 2.5% glutaraldehyde (vol/vol) in 0.1 M sodium cacodylate buffer pH 7.4]. Subsequent cell treatment for TEM was performed as described by Straume et al., (87). Images were taken using a Jeol transmission electron microscope JEM 2100-Plus.

## Whole genome sequencing

Genomic DNA from bacteria (RH14 and ds971) was isolated by using NucleoBond AXG100 columns as described in the included protocol from Macherey–Nagel. Re-sequencing of gDNA from pneumococcal mutants was performed using v3 chemistry on the MiSeq (Illumina) with paired-end reads of 300 bp yielding 90× coverage. Reference-based mapping and mutation analysis were performed using Geneious 8.1.9 and the reference genome of *S. pneumoniae* R6 (Ref: NC_003098.1).

## Hemolysis assay

Sheep blood (Thermo Fisher Scientific) was diluted 1:9 in PBS and centrifuged at 1500 × *g* for 10 minutes. Then the erythrocytes were washed twice in the same volume of PBS before they were re-suspended in PBS corresponding to a 1:9 dilution of the original blood. Aliquots of 990 µL diluted blood were transferred to Eppendorf tubes containing 10 µL viscosin giving a final concentration gradient from 100 to 10 µg/mL. PBS was used as negative control, and a final concentration of 1% (vol/vol) Triton X-100 was used to measure total hemolysis. The samples were incubated at 37°C for 30 minutes. Next, intact erythrocytes were removed from the solution at 1500 × *g* for 10 minutes, and Abs490 of the supernatants was measured.

## ACKNOWLEDGMENTS

We thank the imaging center at the Norwegian University of Life Sciences for electron microscopy analysis. Dr. Morten Skaugen is thanked for valuable help operating the mass spectrometer.

This work was supported by grant nr. 287416 from the Research Council of Norway.

## AUTHOR AFFILIATIONS

[1]Faculty of Chemistry, Biotechnology and Food Science, Norwegian University of Life Sciences, Ås, Norway
[2]Department of Molecular Biology, Norwegian Veterinary Institute, Ås, Norway

## AUTHOR ORCIDs

Anja Ruud Winther  http://orcid.org/0000-0002-0648-4769
Morten Kjos  http://orcid.org/0000-0003-4448-9082
Daniel Straume  http://orcid.org/0000-0002-5222-9275

## FUNDING

| Funder | Grant(s) | Author(s) |
| --- | --- | --- |
| Norges Forskningsråd (Forskningsrådet) | 287416 | Daniel Straume |

## AUTHOR CONTRIBUTIONS

Anja Ruud Winther, Investigation, Methodology, Writing – original draft | Zhian Salehian, Investigation, Methodology | Cathrine Arnason Bøe, Investigation, Methodology, Writing – original draft | Malene Nesdal, Investigation, Methodology | Leiv Sigve Håvarstein, Writing – original draft | Morten Kjos, Investigation, Methodology, Writing – original draft | Daniel Straume, Conceptualization, Investigation, Methodology, Writing – original draft, Writing – review and editing

## ADDITIONAL FILES

The following material is available online.

### Supplemental Material

**Supplemental table and figures (Spectrum00624-24-s0001.pdf).** Table S1; Fig. S1-S5.

### Open Peer Review

**PEER REVIEW HISTORY (review-history.pdf).** An accounting of the reviewer comments and feedback.

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
