## [Reviewer comments · Microbiology Spectrum]

Microbiology Spectrum

Decreased susceptibility to viscosin in *Streptococcus pneumoniae*.

Anja Winther, Zhian Salehian, Cathrine Arnason Bøe, Malene Nesdal, Leiv Sigve Håvarstein, Morten Kjos, and Daniel Straume

Corresponding Author(s): Daniel Straume, Norges miljø- og biovitenskapelige universitet

Review Timeline:

Submission Date:	March 8, 2024
Editorial Decision:	May 14, 2024
Revision Received:	May 24, 2024
Accepted:	May 30, 2024

Editor: Brian Conlon

Reviewer(s): The reviewers have opted to remain anonymous.

Transaction Report:

DOI: <https://doi.org/10.1128/spectrum.00624-24>

Re: Spectrum00624-24 (Acquisition of increased viscosin tolerance by *Streptococcus pneumoniae*.)

Dear Dr. Daniel Straume:

Thank you for the privilege of reviewing your work. Below you will find my comments, instructions from the Spectrum editorial office, and the reviewer comments.

Revision Guidelines

Sincerely,
Brian Conlon
Editor
Microbiology Spectrum

Reviewer #1 (Comments for the Author):

The authors have "rediscovered" viscosin, an antibiotic first discovered in 1950. This is an occupational hazard in the search for new antimicrobial agents. Including a structure of viscosin would be helpful for the reader. The manuscript focuses on the antimicrobial affects of viscosin on *Streptococcus pneumoniae*, and emphasizes sub MIC effects. Genetic changes in a selected mutant showing a modest decrease in susceptibility to viscosin are described. The work seems competently done. A minor amount of English polishing needs to occur.

Reviewer #2 (Comments for the Author):

This manuscript describes a screen for microorganisms that produce antimicrobials with activity against *Streptococcus pneumoniae*. The authors purified a compound with potent bactericidal activity, which they identified as the lipopeptide viscosin. Lipopeptide antibiotics are typically characterized by their ability to form pores in the cytoplasmic membrane. Although, viscosin is not novel, the authors observe that this compound affects cell wall synthesis but not the membrane at sub-MIC levels, indicating that membrane damage may only occur at higher concentrations. The experiments in the manuscript are well designed and the paper is well written. Some minor critiques below.

Line 185. The authors generate a resistant mutant but describe this mutant as having 'increased tolerance'. Tolerance is typically characterized as the ability to withstand lethal doses of drugs without an increased in MIC so this interpretation of the results is confusing. The authors should consider rephrasing this to 'increased resistance'.

Title. The authors should also consider rewording the title based on the comment above.

Line 49: acquired should be acquire.

The resistant mutant has many mutations. To determine which mutation confers resistance to viscosin, the authors could consider introducing these mutations into a clean background. Or at least, acknowledging that many of these mutations may not contribute to the phenotype.

A lipopeptide control such as daptomycin would have been useful in many of these experiments to compare to viscosin.

Response to the Reviewers' comments.

Reviewer #1 (Comments for the Author):

The authors have "rediscovered" viscosin, an antibiotic first discovered in 1950. This is an occupational hazard in the search for new antimicrobial agents. Including a structure of viscosin would be helpful for the reader. The manuscript focuses on the antimicrobial affects of viscosin on *Streptococcus pneumoniae*, and emphasizes sub MIC effects. Genetic changes in a selected mutant showing a modest decrease in susceptibility to viscosin are described. The work seems competently done. A minor amount of English polishing needs to occur.

We thank the reviewer for positive feedback on the manuscript. A structure of viscosin has been included as suggested (Fig. 1c in the revised version). We have carefully read the manuscript and done our best to polish the English. Minor changes to the text are indicated in the "marked-up" file.

Reviewer #2 (Comments for the Author):

This manuscript describes a screen for microorganisms that produce antimicrobials with activity against *Streptococcus pneumoniae*. The authors purified a compound with potent bactericidal activity, which they identified as the lipopeptide viscosin. Lipopeptide antibiotics are typically characterized by their ability to form pores in the cytoplasmic membrane. Although, viscosin is not novel, the authors observe that this compound affects cell wall synthesis but not the membrane at sub-MIC levels, indicating that membrane damage may only occur at higher concentrations. The experiments in the manuscript are well designed and the paper is well written. Some minor critiques below.

Line 185. The authors generate a resistant mutant but describe this mutant as having 'increased tolerance'. Tolerance is typically characterized as the ability to withstand lethal doses of drugs without an increased in MIC so this interpretation of the results is confusing. The authors should consider rephrasing this to 'increased resistance'.

Thanks for clarifying this. We agree with the reviewer that using the word "tolerance" can be confusing. We chose this term based on the relatively small increase in MIC, which we were reluctant to define as a resistant mutant. In the revised version, instead of saying that the mutant has "increased tolerance" we have rephrased this to "decreased susceptibility".

Title. The authors should also consider rewording the title based on the comment above.

*The title has been changed to "Decreased susceptibility to viscosin in *Streptococcus pneumoniae*."*

Line 49: acquired should be acquire.

Corrected.

The resistant mutant has many mutations. To determine which mutation confers resistance to viscosin, the authors could consider introducing these mutations into a clean background. Or at least, acknowledging that many of these mutations may not contribute to the phenotype.

We fully agree with the reviewer that not all the mutations necessarily contribute to increased resistance. During this study we did in fact try to look into this by recreating the mutant by sequentially introducing the mutated genes into a clean background. However, it turned out to be an extremely challenging task mostly because of the essentiality of rpoB and fabK. To replace the native genes with mutated versions, we used the janus system. In theory we can knock out the essential genes with janus if we at the same time express a copy of the native gene ectopically. Then janus can be replaced by the mutated version of rpoB/fabK, and the ectopic copy is finally deleted. What we experienced when working with ropB and fabK (particularly for rpoB) was that the cells were very sensitive to the level of ectopic expression of these genes. For rpoB, we could not find the right expression levels (using different inducer concentrations to drive ectopic expression) that allowed deletion of the native copy. Having all these technical struggles, we decided not to pursue this approach further at the moment. To acknowledge that some of the mutations may not directly contribute to resistance we have written the following in the discussion, line 280-292 “It is also possible that some of the mutations in strain ds971 do not contribute to decreasing viscosin susceptibility, but instead compensate for the fitness cost of mutations in other genes.”

A lipopeptide control such as daptomycin would have been useful in many of these experiments to compare to viscosin.

This is a good point raised by the reviewer. During this work we performed transmission electron microscopy of daptomycin treated cells for comparison with viscosin treatment. The images showed that daptomycin treatment resulted in a somewhat different phenotype compared to viscosin. Daptomycin treated cells were elongated with multiple septa that seemed to have been aborted at an early stage. In addition, the cell wall did not appear less dense, like we observed for viscosin treated cells. We did not include this data in the first version of the manuscript since we reasoned it did not add much to how viscosin works. However, in retrospect we acknowledge that data suggesting that viscosin has a different mode of action than daptomycin may be useful to the readers and have chosen to include the images in the supplemental material as Fig. S4 in the revised version.

*In line 230-232 we have added “We found that daptomycin treatment of *S. pneumoniae* resulted in multiple incomplete septa, but not cell walls of lower density (Fig. S4) as observed for viscosin treated cells, suggesting that viscosin has a different mode of action.”*

Also, since we have previously observed that moenomycin treatment leads to cell walls of lower density (Class A PBPs have a distinct and unique role in the construction of the pneumococcal cell wall - PubMed (nih.gov)) we have included the sentence “The latter has also been reported for pneumococci treated with moenomycin, which inhibits the glycosyl transferase activity of class A PBPs.” in line 222-223.

Re: Spectrum00624-24R1 (Decreased susceptibility to viscosin in *Streptococcus pneumoniae*.)

Dear Dr. Daniel Straume:

Your manuscript has been accepted, and I am forwarding it to the ASM production staff for publication. Your paper will first be checked to make sure all elements meet the technical requirements. ASM staff will contact you if anything needs to be revised before copyediting and production can begin. Otherwise, you will be notified when your proofs are ready to be viewed.

Sincerely,
Brian Conlon
Editor
Microbiology Spectrum